

# Stress management strategies and controlling emotion among Polish nurses and midwives who suffer COVID: cross-sectional study

Barbara Zych[1], Anna Kremska[1], Anna Lewandowska[2] and Małgorzata Nagórska[3]

[1] Institute of Health Sciences, Medical College of Rzeszow University, Rzeszow, Poland
[2] Institute of Healthcare, State School of Technology and Economics, Jaroslaw, Poland
[3] Institute of Medical Sciences, Medical College of Rzeszow University, Rzeszow, Poland

## ABSTRACT

**Introduction**. Stress management strategies and the ability of nursing staff to control their emotions is an important way to reduce emotional tension in a difficult situation.
**Aim**. To identify the dominant stress management strategy and emotion control modality in professionally active nurses and midwives during the COVID-19 pandemic.
**Material and Methods**. A total of 137 nursing personnel from south-eastern Poland were studied. Sociodemographic and clinical data were collected with a questionnaire developed by the authors, stress management was assessed with the Coping Inventory for Stressful Situations (CISS), and control of emotions was examined with the Courtauld Emotional Control Scale (CECS).
**Results**. The most common strategy for stress management strategy among nursing staff was the task-oriented strategy, the least frequently used was the avoidance. Only the avoidance style showed a significant difference in the group of midwives taking the form of seeking social contact ($p = 0.016$). CECS in none of the subcategories showed a significant difference for the profession. It turned out that the longer the time elapsed in nursing staff from having contracted COVID-19, the less often they chose the avoidance oriented coping ($p = 0.022$), and the presence of more post-COVID complications favoured focusing on emotions ($p = 0.016$) and avoidance ($p = 0.005$) in the form of initiating social contacts ($p = 0.003$).
**Conclusions**. The tendency to prefer a maladaptive style of coping with stress and suppression of emotions in nursing staff is associated with the risk of psychosomatic diseases and occupational burnout. The results indicate the necessity of providing interprofessional support combined with learning to "actively cope" with stress at work.

# INTRODUCTION

On March 11, 2020, coronavirus was declared a pandemic (COVID-19) by the World Health Organization (WHO), caused by coronavirus 2 (Sars-Cov-2) causing acute respiratory failure. By the time we began our study, the pandemic had taken a weekly toll of 3 million new cases and 8,000 deaths in Europe alone (*ECDC-WHO, 2022*). In Poland, the first case

Corresponding author
Barbara Zych, ba.zyc@wp.pl

of coronavirus infection was reported on March 4, 2020, and on December 2, 2020, the number of infected people exceeded 1 million, where the average record number of deaths in Poland was 674 people, for 38,036 citizens (*Rozkrut, 2022*). In Poland, by February 2, 2022, 122,826 nurses and midwives were infected with coronavirus, and 289 of them died (*Supreme Chamber of Nurses Midwives, 2022*). This situation acts as an additional stressor and poses a threat to the mental health of Polish nurses and midwives who, despite the sanitary regime, worked with SARS-COV2 patients, putting their own lives at risk (*Pappa et al., 2020*; *Kozina et al., 2022*). In order to prevent the spread of the coronavirus, all countries in the world implemented a strict "anti-COVID" regime, and the protective policies in place included restrictions, isolation, quarantine, and, over time, mass testing and vaccination (*Canan, Murat & Cetin, 2021*). Over time, it became apparent that the COVID-19 pandemic in people around the world caused an increase in anxiety and depressive symptoms (*Pappa et al., 2020*). Health care workers, who worked beyond their means to help an infected population, appeared to be particularly burdened during this time (*Pappa et al., 2020*). Nurses and midwives also found themselves at high risk of contracting the virus, caring for sick patients and testing the public for coronavirus. Shift work causing high periodic overload when another team member became ill, lack of satisfactory pay or lack of recognition from superiors also became a substrate of stress. This traumatic situation contributed to nurses and midwives experiencing strong negative emotions such as fear, anxiety or anger, the suppression of which led to increased or long-lasting emotional tension and lowered immunity (*Pappa et al., 2020*; *Riedel et al., 2021*). The most common disorders that can be caused by excessive stress include cardiovascular disorders, skin diseases, gastrointestinal disorders and depression (*Xie et al., 2022*). Prevention of the negative effects of stress and the ability to cope with it is a key condition for maintaining high performance of professional duties and sustaining job satisfaction (*Wood et al., 2012*; *Zhang et al., 2022*).

In our study, we decided to compare two professional groups of nurses and midwives, which are two different independent medical professions under Polish law. Nurses work with different patients in different specialisations, have a wider choice of workplaces and can care for patients of both sexes, including men. Midwives, on the other hand, focus on caring for women and newborns, and their scope of work is narrower, which limits their access to other hospital wards. According to the researchers, this specific nature of their work may have influenced the level of stress they experienced when exposed to SARS-Cov-2.

The present study aimed to identify the dominant stress management strategy and emotion control method in active nurses and midwives during the COVID-19 pandemic.

## MATERIALS AND METHODS

After obtaining approval from the Bioethics Committee at the Faculty of Medicine of the University of Rzeszow in Rzeszów (Resolution No. 4/04/2021 of the Bioethics Committee of the UR dated April 22, 2021), active nurses and midwives were included in the study. The principles of the Declaration of Helsinki were followed at all stages of conducting the study.

The criteria for inclusion in the study were: having the right to practice as a nurse or midwife and being in a professional relationship with the employer, COVID-19 infection, giving informed consent to participate in the study and signing the consent form. Each respondent signed an informed consent form before completing the questionnaire.

The criteria for exclusion from the study were the lack of a license to practice as a nurse or midwife not having COVID-19 and lack of consent to participate in the study.

This was a descriptive cross-sectional study. The study involved 137 respondents (41 nurses *vs.* 96 midwives) working in medical facilities of various referral levels in south-eastern Poland (Podkarpackie Province). The size of the study population was determined using a research calculator. In our case, the total number of nurses and midwives employed in the year of the study was 15,817, accounting for 5.9% of the total number of nurses and midwives employed in Poland (*Supreme Chamber of Nurses Midwives, 2022*). The maximum determination was 5.0% and the confidence level was 95.0%, which for the study population is at least 143 people. During the research 200 questionnaires were distributed, of which 146 were returned, and the final analysis included 137 correctly completed questionnaires, which is 68.5%. In addition, the main study was preceded by a pilot study, the results of which were not included in the main study data. This was done not only to verify the research tool, but also to subject the data obtained to *post-hoc* analysis. This allowed for the accurate identification of specific differences between the groups of respondents, helped to develop the final research objective, and verified the hypotheses. The study used a self-constructed information questionnaire, the Coping Inventory for Stressful Situations (CISS) and the Courtauld Emotional Control Scale (CECS).

The information questionnaire developed for this study collected sociodemographic and clinical data related to symptoms associated with COVID-19 outbreaks. Study participants were asked about their gender, age, education, work seniority and place of work, subjective feelings of health before the pandemic, clinical symptoms associated with COVID-19 outbreaks and complications after the outbreak.

## Coping Inventory for Stressful Situations

The study used the Polish adaptation of the Coping Inventory for Stressful Situations (CISS) by *Endler & Parker (1990)* and *Strelau et al. (2013)*. The questionnaire consists of 48 statements rated on a five-point Likert scale (1 means never, 2—very rarely, 3—sometimes, 4—often, 5—very often), which ranks scores in three styles of coping with stress: task-oriented coping (TOC), emotion-oriented coping (EOC) and avoidance-oriented coping (AOC) in two variations: distraction (D subscale) and social diversion (SD subscale). The higher the score for a particular style, the more often the person uses the strategies included in it. Correlations between the scales were high and ranged from 0.90 to 0.93. The reliability coefficient for basic scales (TOC, EOC, AOC) was in the range of 0.78−0.90, while for *SC* and *SA* scales it is 0.74−0.90 (*Strelau et al., 2013*).

## Courtauld Emotional Control Scale

Emotional control was measured using the Courtauld Emotional Control Scale (CECS), adapted in Poland by *Jurczynski (2012)* developed by Watson and Greer. The scale assesses
the extent to which a person reports suppression as an emotion regulation strategy in difficult situations. It consists of 21 statements on the disclosure of three basic emotions, *i.e.,* anger, depression and anxiety. The Polish version of the CECS, compared to the original, includes a change from two negative statements to positive ones, thus reducing the number of errors in responses to a statement containing a double negation (statement No. 2 on the anger scale and No. 1 on the depression scale). In the Polish version, the Cronbach's alpha coefficient for anger control is 0.80; depression 0.77; anxiety 0.78 and for the total emotion control index 0.87, and was similar to the coefficients in the original version (*Jurczynski, 2012*).

## Statistical analysis

Statistical analysis of the collected material was performed using Statistica 13.3, a StatSoft package. For sociodemographic variables, frequencies (number, percentage) are indicated. The correlation of two variables with at least ordinal data type was determined using Spearman's rank correlation coefficient. Analysis of variables having qualitative data type was carried out using Pearson's chi-square test. The Pearson's chi-square test was used to analyze categorical data: health status before COVID-19 infection, the course of disease, the infection method diagnosis, symptoms spectrum, complications spectrum after infection.

The Mann–Whitney U-Test was used to assess continuous variables: stress management, emotional control score.

Spearman's rank correlation test was used to assessment of correlation between rank values and continuous variables: Time since disease, severity of the disease course, number of symptom, number of complications, TOC, EOC, AOC, D, SD.

The level of statistical significance was adopted at $p < 0.05$.

## RESULTS

The study included 137 nurses and midwives (nurses: 41 *vs.* -midwives: 96) who underwent COVID-19. Women accounted for 96.4% of the respondents, and men for 3.6%. The average age of all respondents was 37.82 years $\pm$ 12.31 years (nurses: 39.15 $\pm$ 9.47 years; midwives 37.26 $\pm$ 13.34 years). The average work seniority of nurses was 13.51 years and midwives 11.76 ($p = 0.442$).

Most of the subjects lived in rural areas (52.6%), and they worked in a hospital (76.6%) with a level II reference (55.3%) (Table 1).

The nurses and midwives described their subjective health status before contracting COVID-19 as good (nurses: 48.8% vsmidwives: 38.5%) or very good (nurses: 41.5% *vs.* midwives: 39.6%), indicating a mild course of the disease (82.5%). The time since contracting COVID-19 among all subjects was on average 4.78 months $\pm$ 2.51 months ($p = 0.773$) and was confirmed by genetic testing (50.4%) (Table 2). SARS-Cov-2 infection was confirmed by available tests, *i.e.,* in 16.6% of respondents on the basis of a serological test, in 50.4% a genetic test, in 12.4% an antigen test, and in 22.6% the infection was confirmed only on the basis of symptoms. None of the subjects had been hospitalized for COVID-19 ($p = 1,000$).

**Table 1  Sociodemographical data.**

| Variables | | Nurses (n = 41) n, (%) | Midwives (n = 96) n, (%) | Total n, (%) |
|---|---|---|---|---|
| Gender | Woman | 36, (87.8) | 96, (100.0) | 132, (96.4) |
| | Man | 5, (12.2) | 0, (0.0) | 5, (3.6) |
| Place of residence | City | 15, (36.6) | 50, (52.1) | 65, (47.4) |
| | Village | 26, (63.4) | 46, (47.9) | 72, (52.6) |
| Place of work | Hospital | 36, (87.8) | 69, (71.9) | 105, (76,6) |
| | Clinic | 5, (12.2) | 27, (28.1) | 32, (23.4) |
| Hospital's degree of reference | I | 22, (53.7) | 37, (40.7) | 59, (44.7) |
| | II | 19, (46.3) | 45, (49.5) | 64, (48.5) |
| | III | 0, (0.0) | 9, (9.9) | 9, (6.8) |

**Table 2  Subjective assessment of health status before and after COVID-19, and way of infection confirmation.**

| Variables | | Nurses (n = 41) n, (%) | Midwives (n = 96) n, (%) | Total n, (%) | p |
|---|---|---|---|---|---|
| Subjective COVID-19 pre-infection health assessment | Bad | 0, (0.0) | 0, (0.0) | 0, (0.0) | |
| | Average | 1, (2.4) | 6, (6.3) | 7, (5.1) | |
| | Moderate | 3, (7.3) | 15, (15.6) | 18, (13.1) | 0.380 |
| | Good | 20, (48.8) | 37, (38.5) | 57, (41.6) | |
| | Very good | 17, (41.5) | 38, (39.6) | 55, (40.2) | |
| Subjective assessment of the course of one's own illness | Asymptomatic | 2, (4.9) | 7, (7.3) | 9, (6,6) | |
| | Mild | 34, (82.9) | 79, (82.3) | 113, (82.5) | 0.844 |
| | Heavy | 5, (12.2) | 10, (10.4) | 15, (11.0) | |
| Test with which Sars-CoV-2 infection was confirmed | Serological | 6, (14.6) | 14, (14.6) | 20, (14.6) | |
| | Genetic | 18, (43.9) | 51, (53.1) | 69, (50.4) | |
| | Antigen | 9, (22.0) | 8, (8.3) | 17, (12.4) | 0.169 |
| | Infection confirmed based on symptoms alone | 8, (19.5) | 23, (24.0) | 31, (22.6) | |

The symptoms of coronavirus in the group of nurses and midwives studied were similar. No statistically significant differences were shown in this regard ($p > 0.05$). The most frequently indicated symptoms were headache (63.5%), olfactory disturbance (62.8%), taste disturbance (58.4%), cough (46.0%), myalgia (45.3%), fever (41.6%) and loss of appetite (41.6%) (Table 3).

The complications experienced by nursing staff after contracting coronavirus did not differ significantly. The most common were fatigue (61.3%), difficulty concentrating (35.0%) and headaches (21.9%). The only significant difference after COVID-19 outbreak was observed in loss of smell in midwives ($p = 0.031$) (Table 4).

Using the CISS scale, we assessed nurses' and midwives' personal resources during the pandemic. Of the three main stress coping strategies, the task-oriented coping was the most strongly expressed (average 56.90 points). This was followed by avoidance-oriented coping

**Table 3  COVID-19 symptoms in study subjects.**

| COVID-19 symptoms | Nurses (n = 41) n, (%) | Midwives (n = 96) n, (%) | Total n, (%) | p |
|---|---|---|---|---|
| Cough | 19, (46.3) | 44, (45.8) | 63, (46.0) | 0.816 |
| Fever | 15, (36.6) | 42, (43.8) | 57, (41.6) | 0.436 |
| Shortness of breath/difficulty breathing | 9, (22.0) | 22, (22.9) | 31, (22.6) | 0.902 |
| Olfactory disorders | 24, (58.5) | 62, (64.6) | 86, (62.8) | 0.721 |
| Taste disorders | 23, (56.1) | 57, (59.4) | 80, (58.4) | 0.721 |
| Loss of appetite | 14, (34.2) | 43, (44.8) | 57, (41.6) | 0.247 |
| Diarrhea | 2, (4.9) | 8, (8.3) | 10, (7.3) | 0.476 |
| Shivers | 13, (31.7) | 35, (36.5) | 48, (35.0) | 0.593 |
| Myalgia (muscle pain) | 15, (36.6) | 47, (49.0) | 62, (45.3) | 0.182 |
| Nausea and vomiting | 3, (7.3) | 14, (14.6) | 17, (12.4) | 0.237 |
| Headache | 23, (56.1) | 64, (66.7) | 87, (63.5) | 0.239 |
| Sore throat | 7, (17.1) | 26, (27.1) | 33, (24.1) | 0.209 |
| Conjunctivitis | 2, (4.9) | 4, (4.2) | 6, (4.4) | 0.852 |
| Irritability | 4, (9.8) | 12, (12.5) | 16, (11.7) | 0.647 |
| Confusion | 0, (0.0) | 7, (7.3) | 7, (5.1) | 0.076 |
| Memory problems | 8, (19.5) | 22, (22.9) | 30, (21.9) | 0.659 |
| Skin rash | 1, (2.4) | 3, (3.1) | 4, (2.9) | 0.827 |
| Discoloration of fingers and toes | 1, (2.4) | 0, (0.0) | 1, (0.7) | 0.124 |
| Anxiety | 8, (19.5) | 23, (24.0) | 31, (22.6) | 0.568 |
| Depressive states | 1, (2.4) | 7, (7.3) | 8, (5.8) | 0.267 |
| Sleep disorders | 9, (22.0) | 31, (32.3) | 40, (29.2) | 0.222 |
| Neurological disorders | 0, (0.0) | 1, (1.0) | 1, (0.7) | 0.511 |

(average 45.31 points) and emotion-oriented coping (average 43.53 points). Distraction was rated at 19.67 points, and social diversion at 17.26 points (Table 5). The results obtained by nurses and midwives in all the strategies of coping with stress considered were comparable. An avoidant style of coping with stress, focusing on seeking social contacts, was identified in our study group ($p = 0.016$). In the group of midwives, the avoidance strategy used in coping with stressful situations proved to be significant, which indicates/may indicate the fact of temporary distraction from the problem (PKT), while stress remains.

Overall, the respondents scored an average of 49.06 points on the CECS scale. The scores in the two study groups did not differ significantly from each other ($p = 0.291$). There were also no confirmed significant differences in any of the three subcategories of the CECS scale, *i.e.,* anger, depression and anxiety, even though the differences in anger ($p = 0.097$) and anxiety ($p = 0.090$) scale scores were close to the threshold of significance. Anger and anxiety, were more strongly expressed in the nurses' group than in the midwives' group (Table 6).

In Table 7, quantitative data on stress coping styles (TOC; EOC, AOC, D, SD) and the emotion control scale (CECS) were analysed based on the time elapsed since the onset of the disease (numerical data in months), subjective assessment of the course of the

**Table 4  The persisting symptoms that occurred after COVID-19 in the study group.**

| The persisting symptoms after COVID-19 | Nurses (n = 41) n, (%) | Midwives (n = 96) n, (%) | Total n, (%) | p |
|---|---|---|---|---|
| Fatigue | 22, (53.7) | 62, (54.6) | 84, (61.3) | 0.229 |
| Shortness of breath | 2, (4.9) | 13, (13.5) | 15, (11.0) | 0.136 |
| Cough | 4, (9.8) | 14, (14.6) | 18, (13.1) | 0.444 |
| Joint pain | 3, (7.3) | 14, (14.6) | 17, (12.4) | 0.237 |
| Chest pain | 0, (0.0) | 3, (3.1) | 3, (2.2) | 0.252 |
| Lower mood | 8, (19.5) | 19, (19.8) | 27, (19.7) | 0.969 |
| Muscle pains | 2, (4.9) | 7, (7.3) | 9, (6.6) | 0.602 |
| Headaches | 5, (12.2) | 25, (26.0) | 30, (21.9) | 0.072 |
| Dizziness | 2, (4.9) | 13, (13.5) | 15, (11.0) | 0.136 |
| Recurrent fever | 0, (0.0) | 1, (1.0) | 1, (0.7) | 0.511 |
| Heart palpitations | 7, (17.1) | 15, (15.6) | 22, (16.1) | 0.832 |
| Loss of smell | 0, (0.0) | 10, (10.4) | 10, (7.3) | 0.031 |
| Taste disorders | 2, (4.9) | 8, (8.3) | 10, (7.3) | 0.476 |
| Short-term memory loss | 5, (12.2) | 11, (11.5) | 16, (11.7) | 0.902 |
| Difficulties with concentration | 15, (36.6) | 33, (33.4) | 48, (35.0) | 0.804 |
| More frequent infections | 0, (0.0) | 4, (4.2) | 4, (2.9) | 0.184 |

**Notes.**
$\chi_2$, Pearson's chi-square test value; $p$, test probability ratio.

**Table 5  Stress management strategies of nurses and midwives.**

| CISS | Basic descriptive statistics | | | | | | p |
|---|---|---|---|---|---|---|---|
| | Total | | Nurses | | Midwives | | |
| | X ± SD | Min.–Max. | X ± SD | Min.–Max. | X ± SD | Min.–Max. | |
| TOC [16–80 pts.] | 56.90 ± 8.59 | 16.00–79.00 | 54.54 ± 9.37 | 16.00–70.00 | 57.91 ± 8.07 | 39.00–79.00 | 0.053 |
| EOC [16–80 pts.] | 43.53 ± 9.49 | 23.00–68.00 | 42.85 ± 9.79 | 27.00–64.00 | 43.81 ± 9.40 | 23.00–68.00 | 0.660 |
| AOC [16–80 pts.] | 45.31 ± 7.98 | 45.00–66.00 | 43.63 ± 7.58 | 43.00–57.00 | 46.03 ± 8.08 | 46.00–66.00 | 0.174 |
| D [8–40 pts.] | 19.67 ± 5.42 | 7.00–32.00 | 19.71 ± 5.66 | 9.00–28.00 | 19.66 ± 5.35 | 7.00–32.00 | 0.836 |
| SD [5–25 pts.] | 17.26 ± 3.54 | 9.00–25.00 | 16.05 ± 3.15 | 9.00–22.00 | 17.77 ± 3.59 | 9.00–25.00 | 0.016 |

**Notes.**
TOC, task-oriented coping style; EOC, emotion-oriented coping; AOC, avoidance-oriented coping; D, distraction; SD, social diversion; Z-score of Mann–Whitney U-test; p-test probability level.

disease (taking into account the course 0-asymptomatic; 1-mild and 2-severe) and the number of symptoms and complications per respondent. Among the total study group, one negative correlation was significant—the longer the time since the illness, the less often the study subjects chose an avoidance-oriented coping with stress. There were three positive correlations—the higher the number of post-covid complications the subjects had,

**Table 6  Courtauld Emotional Control Scale scores of the nurses and midwives surveyed.**

| CECS | Basic descriptive statistics | | | | | | p |
|---|---|---|---|---|---|---|---|
| | Total | | Nurses | | Midwives | | |
| | X ± SD | Min.–Max. | X ± SD | Min.–Max. | X ± SD | Min.–Max. | |
| CECS total | 49.06 ± 10.12 | 22.00–80.00 | 50.12 ± 11.04 | 22.00–80.00 | 48.60 ± 9.72 | 29.00–77.00 | 0.291 |
| CECS anger | 16.20 ± 4.22 | 7.00–28.00 | 16.83 ± 3.97 | 8.00–25.00 | 15.93 ± 4.31 | 7.00–28.00 | 0.097 |
| CECS depression | 16.39 ± 4.04 | 7.00–27.00 | 16.12 ± 4.54 | 7.00–27.00 | 16.50 ± 3.82 | 7.00–24.00 | 0.789 |
| CECS anxiety | 16.47 ± 4.19 | 7.00–28.00 | 17.15 ± 4.17 | 7.00–28.00 | 16.18 ± 4.19 | 7.00–27.00 | 0.090 |

**Table 7  Assessment of relationships between selected variables in nurses and midwives combined.**

| Variables | Time since illness | | Severity of the course of the disease | | Number of symptoms | | Number of complications | |
|---|---|---|---|---|---|---|---|---|
| | R | p | R | p | R | p | R | p |
| TOC | 0,04 | 0,654 | −0,08 | 0,326 | −0,03 | 0,711 | −0,07 | 0,387 |
| EOC | −0,12 | 0,153 | −0,09 | 0,279 | 0,16 | 0,066 | 0,21 | 0,016 |
| AOC | −0,20 | 0,022 | 0,06 | 0,475 | 0,08 | 0,372 | 0,24 | 0,005 |
| D | −0,15 | 0,091 | 0,03 | 0,686 | 0,02 | 0,839 | 0,25 | 0,003 |
| SD | −0,05 | 0,533 | 0,04 | 0,649 | 0,06 | 0,502 | 0,02 | 0,831 |
| CECS total | 0,10 | 0,258 | 0,12 | 0,160 | 0,02 | 0,851 | 0,08 | 0,335 |
| CECS anger | 0,01 | 0,885 | 0,16 | 0,059 | 0,07 | 0,421 | 0,12 | 0,180 |
| CECS depression | 0,03 | 0,754 | 0,14 | 0,106 | 0,01 | 0,887 | 0,02 | 0,775 |
| CECS anxiety | 0,17 | 0,051 | 0,02 | 0,831 | −0,06 | 0,453 | 0,03 | 0,697 |

**Notes.**
TOC, task-oriented coping style; EOC, emotion-oriented coping; AOC, avoidance-oriented coping; D, Distraction; SD, Social Diversion; R- score in Spearman's rank correlation test; *p*-test probability level.

the more often they responded to stress with an emotion-oriented and avoidance-oriented coping, distracting (Table 7).

# DISCUSSION

COVID-19 caused many people not to expect the magnitude of the epidemic, and the accompanying permanent stress of infection and possible death caused negative emotions. In our study, the personal resources of nursing staff, analyzing their stress management strategy and control of emotions in a pandemic showed that the dominant style was EOC and AOC. Both of these styles are considered maladaptive, leading to a lack of realism in action. As the literature indicates, over time this situation can lead to anxiety disorders, addiction, psychosis, post-traumatic stress disorder (*Chen et al., 2005*; *Denning et al., 2021*), depression, and suicidal thoughts (*Tracy et al., 2020*; *Omar, Amer & Abdelmaksoud, 2023*). Many studies have shown that healthcare professionals directly involved in diagnosing, treating, or providing nursing care to patients with suspected or confirmed COVID-19 were more likely to develop psychological symptoms (*Lai et al., 2020*). These professionals are more likely to have sleep disorders (*Ghahramani et al., 2023*), symptoms of anxiety, depression, post-traumatic stress disorder (PTSD), and burnout (*Li*

*et al., 2021*). In addition, insomnia, clinical symptoms of depression, anxiety, and PTSD have been found to be common in all healthcare workers during the COVID-19 pandemic (*Ghahramani et al., 2023*; *Lee et al., 2023*).

We also observed that as the time passed after experiencing COVID-19, the nurses and midwives studied were more likely to refrain from engaging in distraction behavior (AOC). But already, the occurrence of more post-COVID complications in them resulted in a focus on emotions and avoidance, accompanied by wishful thinking and distraction behaviors (*Chen et al., 2005*; *Jurczynski, 2012*). In this context, the results of our study raise some concerns. The tendency to cope with stress using a non-adaptive style with a SD subtype as less stressful suggests that with the multiplicity of post-covid symptoms, the tendency to avoid the underlying problem by seeking substitute contacts increases. In this sense, interpersonal contacts do not provide the utilized support to solve the problem, but only serve to reduce tension and serve to ''forget'' the difficult situation. By virtue of the nurses' and midwives' profession, the ability to control emotions is expected, because the health, life and safety of patients depend on how effectively these two professions handle stressful situations (*Chen et al., 2005*; *Góes et al., 2020*; *Denning et al., 2021*). Lack of stress management skills of nursing staff will cause occupational burnout syndrome, developing with the dynamic interaction of the person and the accompanying situation (*Maslach, Schaufeli & Leiter, 2001*). An important role in this situation is played by the nurse's and midwive's personality, which can be expressed quite differently, depending on the adaptation to stress. From the five-factor model of personality (neuroticism, extraversion, openness to experience, agreeableness and conscientiousness), neuroticism is characterized by susceptibility to experiencing negative emotions such as fear, confusion, anger, guilt and sensitivity to psychological stress (*Afshar et al., 2015*; *Rahman & Plummer, 2020*). In this case, suppression of emotions and failure to express them leads to an increase in long-term emotional tension and anxiety, which up to 40% of nurses struggle with and is more severe in younger women (*Pappa et al., 2020*; *Koontalay et al., 2021*; *Rouhbakhsh et al., 2022*; *Omar, Amer & Abdelmaksoud, 2023*). In our study, we observed stronger expressed anger and fear in the group of nurses studied (anger by 0.90 points and anxiety by 0.97 points in relation to midwives, $p = 0.097$ and $p = 0.789$, respectively), but these results were statistically insignificant. In our opinion, this situation may be related to the 12-hour shift work beyond their strength and the diverse group of patients, which proved even more stressful in the pandemic due to the complexity of the disease development and its possible consequences related to the nurse's transmission of the infection to their families (*Rahman & Plummer, 2020*; *Koontalay et al., 2021*). Similar observations were made in a group of nurses from China (*Huang et al., 2020*), Spain (*Lorente, Vera & Peiró, 2021*) Saudi Arabia (*Moussa et al., 2021*), Egypt (*Góes et al., 2020*) and Malaysia (*Ismail et al., 2023*). It was even demonstrated that COVID-19 stress significantly negatively correlated with readiness and willingness to undertake nursing care for infected patients (*Labrague, 2021*). The work of a nurse and midwife is multifaceted and requires the person performing it to have certain predispositions and skills, so it can be thought that the entire professional group is potentially exposed to a stressor in the form of strong negative feelings, which, if chronically sustained, can cause psychosomatic disorders (*Bohlken et al., 2020*; *Cheung,*

*Fong & Bressington, 2020*; *Chew et al., 2020*). Unexpressed emotions, over time, can cause neurotic disorders and psychosomatic diseases. Supressed anger and anxiety can be the cause of some diseases, causing increased blood pressure, accelerated heart rate and increased risk of cardiovascular disease (*Ehrenthal et al., 2010*; *Koontalay et al., 2021*).

### Limitations of the study

Our study has some limitations. The sample may not be representative of the broader population of Polish nurses and midwives, as the study was cross-sectional and was conducted during the lowering of the third wave of the pandemic. Another limitation in our work was the lack of assessment of the health of respondents after COVID-19 infection. Undoubtedly, further multidimensional studies examining the effects of the pandemic and its impact on the actions taken by Polish nurses and midwives are needed.

## CONCLUSIONS

The specific nature of the work of nursing personnel, in addition to performing medical procedures, is to interact with other people and accompany them in difficult situations, which, during the pandemic, turned out to be work in arduous conditions and under the influence of strong emotions. Thus, stress is a constant companion of the work performed by the nurse and midwife. Thus, coping with stress in a nurse's work is an important adaptive skill, in which the more passive and evasive strategies she uses, the more often she will be exposed to psychosomatic diseases and occupational burnout syndrome. Our study points to the need to prepare personalized training for nurses and midwives so that while on duty they know how to take care of themselves and the team they work in, identifying the first symptoms of professional burnout, and learn to focus on adaptive coping strategies to be able to cope with the anxiety of the next difficult situation.

## ACKNOWLEDGEMENTS

We would like to thank the participants in our work for their valuable data and the reviewers for their helpful comments and suggestions to improve our study.

### Funding
The authors received no funding for this work.

### Competing Interests
The authors declare there are no competing interests.

### Author Contributions
- Barbara Zych conceived and designed the experiments, performed the experiments, analyzed the data, prepared figures and/or tables, authored or reviewed drafts of the article, obtaining financing, and approved the final draft.

- Anna Kremska performed the experiments, analyzed the data, prepared figures and/or tables, and approved the final draft.
- Anna Lewandowska analyzed the data, prepared figures and/or tables, and approved the final draft.
- Małgorzata Nagórska analyzed the data, prepared figures and/or tables, and approved the final draft.

### Ethics

The following information was supplied relating to ethical approvals (*i.e.*, approving body and any reference numbers):

Resolution of the Bioethics Committee (Resolution No. 4=04=2021) of the UR dated April 22, 2021.

### Data Availability

The raw measurements are available in the Supplemental File.

### Supplemental Information

Supplemental information for this article can be found online at http://dx.doi.org/10.7717/peerj.19816#supplemental-information.

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
