# Peer review of "Stress management strategies and controlling emotion among Polish nurses and midwives who suffer COVID: cross-sectional study"

_PeerJ, doi:10.7717/peerj.19816_

## Round 0.1 · original submission · Major Revisions

My suggested changes and reviewer comments are shown below and on your article 'Overview' screen.

Please address these changes and resubmit. Although not a hard deadline please try to submit your revision within the next 30 days.

Reviewer 1 ·

Basic reporting

No comment

Experimental design

Line 29. The high mortality among nurses due to COVID in Poland, 289 in 2022, is surprising. This data should be considered as particularly high in comparison with other European countries?. This should be evaluated.
Lines 54-57 Since this was a randomized method, it would be advisable to describe how the recruitment method was used: was an invitation to take the tests sent to all nurses in the southeast region of the country? was it done face-to-face? how was it randomized? from what complete database was the starting point for randomization? were only nurses with symptoms and/or diagnosis of COVID taken into account?
Lines 60-62. It would be useful to specify the clinical units where the participating nurses mainly worked. Primary care is not the same as working in ICU or hospital emergency departments. It is discussed further in line 102 but the typology of the clinical units where nurses and midwives work are not detailed either.
Line 102. Were there no nurses who did not have COVID? Did all the nurses who took the tests must have had COVID to be included in the study? Perhaps this explains why the sample is so low?.
If all the participants in the study suffered from COVID as we are seeing at this point in the article, perhaps it would be interesting to change the orientation of the study from the very moment the title was formulated. Suggestion: “Stress management strategies and controlling emotion among Polish nurses and midwives who suffer COVID: a randomized trial”. It is interesting to see how these professionals behave in the management and control of their emotions when they have already suffered the disease in a context, as it seems to be, of high professional mortality.
Discussion Line. If the point of view that I have been maintaining on the peculiarity of this study is accepted, it would be interesting to allude to whether it could be the case that those professionals who have suffered COVID-19 respond by managing stress and emotions differently from those professionals who have not suffered it during the pandemic.

Validity of the findings

Lines 213 and 216. I believe that it is not convenient to introduce bibliographic citations in the Conclusions section because the discussion with other authors has already been left aside and the conclusions are those of the research team and theirs alone.

Reviewer 2 ·

Basic reporting

I would like to thank the authors for their work.
This is an interesting paper, which aims to to identify the dominant stress management strategy and emotion control method in active nurses and midwives during the COVID-19 pandemic.
Before the publication, some improvements can be done.
The manuscript is written in good English, but I suggest having a native speaker review it to further improve it.
I suggest standardizing the spelling of the term “Covid-19” throughout the text to “COVID-19”
The references are up-to-date.
The background can be improved:
- Lines 33-34: "Over time, it became apparent that the Covid-19 pandemic in people around the world caused an increase in anxiety and depressive symptoms". This sentence needs a reference.
- Lines 38-42: "Shift work causing high periodic overload when another team member became ill, lack of satisfactory pay or lack of recognition from superiors also became a substrate of stress. This traumatic situation contributed to nurses and midwives experiencing strong negative emotions such as fear, anxiety or anger, the suppression of which led to increased or long-lasting emotional tension and lowered immunity.". These sentences need one or more references.
- Lines 44-46 : "Prevention of the negative effects of stress and the ability to cope with it is a key condition for maintaining high performance of professional duties and sustaining job satisfaction." This sentence needs a reference. Nevertheless, I would try to explore these aspects further, to strengthen the background.

Experimental design

This section needs many improvements.
First of all, the design used is not that of a randomized trial; as specified in the text this is a cross-sectional study, so it represents an observational study. I would modify the title of the manuscript accordingly.
Instead of “Design” I would call the chapter “Materials and methods”
Lines 57-58: "Participation in the study was equivalent to signing the informed consent Form."
Lines 60-62: "The study included 137 randomly selected nurses (41 nurses vs. 96 midwives) working in
medical institutions of varying levels of reference in south-eastern Poland (Podkarpackie province)."
I would make it clearer that randomization refers to the sampling procedure. Nevertheless, this part should be deepened. How was the recruitment conducted? What was the total population from which the sample was drawn?

Validity of the findings

Some data reported in the results are unclear, e.g. “The average work seniority of all the subjects was 13.51 years ± 11.76 years (p= 0.442)”. What is this p-value referred to? This aspect recurs in the following results. Instead of Group 1 and Group 2, I would refer to the two groups as nurses and midwives to simplify the reading.
The discussion is consistent with the results and not speculative.
I would move the limitations of the study before the conclusions.

Reviewer 3 ·

Basic reporting

The manuscript studies an important topic about stress management strategies and emotional control among nurses and midwives during the COVID-19 pandemic. However, this study has several fundamental issues regarding study design and reporting clarity. The most critical issue is that while the title claims this is "a randomized trial," the methodology and results clearly indicate this is a cross-sectional observational study. There is no description of randomization process, intervention groups, control groups, or random allocation of participants - this study simply collected data from 137 nurses and midwives at a single time point. The title should be revised to accurately reflect the study design, for example: "Stress management strategies and emotional control in Polish nurses and midwives during pandemic: a cross-sectional study".

Experimental design

1. The sampling method is unclear. Line 60-62 mentions "random selection" but provides no details about the randomization process. If participants were enrolled based on convenience/availability, this should be stated as convenience sampling.

2. There are inconsistent terminology when referring to study groups - sometimes all participants are called "nurses" while elsewhere they are separated into Group I (nurses) and Group II (midwives)
This needs to be standardized throughout for clarity

3. Table 2 reports 31 participants without COVID testing, yet Tables 3-4 use total sample sizes of N=41 and N=96 when reporting COVID complications. If all participants had COVID-19, this should be clearly stated in methods. Need explanation of how COVID-19 status was determined for the 31 participants without test results. Inclusion criteria need to clarify if prior COVID-19 infection was required for participation

Validity of the findings

The validity of the findings in this cross-sectional study is challenged by several major methodological concerns. The observed differences between Group I (nurses) and Group II (midwives) could be largely attributable to uncontrolled confounding factors rather than true group differences. The authors did not adjust for potential confounders in baseline characteristics such as age, work experience, hospital level, and workplace conditions - variables that could substantially explain the observed group differences.

---

## Round 0.2 · Minor Revisions

Thank you for responding to the reviewer comments. I have a few additional items for consideration. Could you please respond to the below:

- In your introduction section, please outline why you chose to compare nurses to midwives - i.e. why would you expect these groups to be different and what is your aim in analysing these two groups separately.
Please state how many nurses were invited to participate in your study, to determine a respondent rate. E.g. a total of 4000 individuals (2000 nurses and 2000 midwives) from three hospitals and two clinics were invited to participate in this study, with 137 consenting to participate (3.4% response rate).

- You have stated that you were required to enrol at least 143 people, but only included 137. Can you please comment on this?

- For your statistical analysis section, it is usual to start with how your baseline characteristics were presented e.g. as mean and SD or median and IQR. Please also state what variables were being correlated in the Spearman's correlation testing. For qualitative variables, do you mean 'categorical' variables for those analysed using the chi square test? How were continuous variables analysed?

- Table 2 is labelled as 'Subjective assessment of health status before and after COVID-19, clinically confirmed' - was any assessment undertaken of health status after COVID-19 infection? This data does not seem to be presented anywhere.

- For the text relating to Table 5, there is no discussion of the differences between nurses and midwives, yet this is how the table is presented. It would be useful to highlight any differences (or lack of) in the accompanying text.

-For tables 5 and 6, it would more usefully presented with nurses and midwives as columns at the top, rather than repeated in rows for every score.

- in Table 7 and in the text, please outline how you determined the variables to be correlated with each of the scores.

- Please provide a copy of the ‘subjective’ questionnaire as a supplemental file.

- The consent form seems to indicate that this is part of a larger study, that extends beyond healthcare professionals. Please provide information concerning this in the text. You have stated that this study was preceded by a post hoc study, but it is not clear what this means. Please re-phrase and provide more context for this.

Reviewer 1 ·

Basic reporting

The authors have satisfactorily responded to the considerations made in the first review. I therefore consider that it can be published without further modification.

Experimental design

No further modifications needed

Validity of the findings

No further modifications needed

Reviewer 3 ·

Basic reporting

Thank you for inviting me to review this manuscript again, the authors have addressed all my previous comments regarding basic reporting.

Experimental design

Thank you for inviting me to review this manuscript again, the authors have addressed all my previous comments regarding experimental design.

Validity of the findings

Thank you for inviting me to review this manuscript again, the authors have addressed all my previous comments regarding validity of the findings.

---

## Round 0.3 · Minor Revisions

Thank you for resubmitting your manuscript and response to my comments. I have some additional comments for consideration. Please address the following:

1. In the section you have added concerning the number of questionnaire respondents, you have stated that you took into account withdrawals and refusals (page 4, line 88). Please add in the number of withdrawals and refusals to reflect the total number of participants that were invited to take part in the study.

2. For Table 1, can you add in further sociodemographic data such as age and sex.

3. As discussed in the last review, please start your statistical analysis section with how baseline sociodemographic details are presented.

4. In Table 2, you refer to health status before and after COVID-19 infection. However, the Table (or accompanying text) does not have any information about health status after COVID-19 infection (it includes data on pre infection and perceived severity of acute infection, but does not include any assessment on post infection health status). Please add this in to the table and text.

5. In Table 4 and accompanying text, you refer to complications post COVID-19. These are persisting symptoms, not complications and I think this should be changed.

6. Finally, in your discussion, you state that anger and anxiety were more strongly expressed in female nurses (page 11, line 254) but there is no data in your results to support this. Please add this in.

---

## Round 0.4 · accepted · Accept

Thank you for addressing my comments. Your manuscript is now ready for publication. Congratulations.